

# Extracting subleading corrections in entanglement entropy at quantum phase transitions

Menghan Song[1°], Jiarui Zhao[1°], Zi Yang Meng[1★], Cenke Xu[2†] and Meng Cheng[3‡]

**1** Department of Physics and HKU-UCAS Joint Institute of Theoretical and
Computational Physics, The University of Hong Kong, Pokfulam Road, Hong Kong SAR, China
**2** Department of Physics, University of California, Santa Barbara, CA 93106
**3** Department of Physics, Yale University, New Haven, Connecticut 06511-8499, USA

★ zymeng@hku.hk , † xucenke@ucsb.edu , ‡ m.cheng@yale.edu

## Abstract

We systematically investigate the finite size scaling behavior of the Rényi entanglement entropy (EE) of several representative 2d quantum many-body systems between a subregion and its complement, with smooth boundaries as well as boundaries with corners. In order to reveal the subleading correction, we investigate the quantity "subtracted EE" $S^s(l) = S(2l) - 2S(l)$ for each model, which is designed to cancel out the leading perimeter law. We find that (1) for a spin-1/2 model on a 2d square lattice whose ground state is the Neel order, the coefficient of the logarithmic correction to the perimeter law is consistent with the prediction based on the Goldstone modes; (2) for the $(2+1)d$ O(3) Wilson-Fisher quantum critical point (QCP), realized with the bilayer antiferromagnetic Heisenberg model, a logarithmic subleading correction exists when there is sharp corner of the subregion, but for subregion with a smooth boundary our data suggests the absence of the logarithmic correction to the best of our efforts; (3) for the $(2+1)d$ SU(2) J-$Q_2$ and J-$Q_3$ model for the deconfined quantum critical point (DQCP), we find a logarithmic correction for the EE *even with smooth boundary*.

## Contents

° These authors contributed equally to the development of this work.

# 1 Introduction

The (Rényi) entanglement entropy between a subregion and its complement encodes universal information of the infrared physics of gapless systems, and gapped systems with topological order. The most well-known example is the leading contribution to the entanglement entropy for a $(1+1)d$ conformal field theory (CFT), which scales logarithmically with the size of the subregion. The coefficient of the logarithmic scaling is proportional to the central charge of the CFT [1,2]. In higher-dimensional CFTs, the EE is dominated by the leading non-universal perimeter law scaling, and it is the subleading terms that may encode the universal information [3,4]. For example, in $(2+1)d$ when the boundary of a subregion $A$ has corners with angles $\alpha_i$, we expect

$$S_A(l) = al - s \ln \frac{l}{\epsilon} + O(1/l), \tag{1}$$

where $\epsilon$ is some short-distance cutoff, and the logarithmic correction coefficient $s$ is a sum of contribution from each corner: $s = \sum_i s(\alpha_i)$. While the complete form of the function $s(\alpha)$ is not known analytically in general, it was shown that for $\alpha$ close to $\pi$, $s(\alpha) \sim \sigma(\pi - \alpha)^2$ and $\sigma$ is proportional to the stress tensor central charge of the CFT [4,5]. Thus, in a CFT, the logarithmic correction is expected to vanish when there are no sharp corners i.e. $\alpha = \pi$.

The EE can also have a universal subleading contribution in an ordered phase that spontaneously breaks a continuous symmetry, i.e., its low energy physics is dominated by the Goldstone modes [6,7]. The Goldstone modes also lead to a logarithmic subleading contribution to the EE, but the logarithmic term receives a contribution from both the corners and the smooth boundary, which is a sharp contrast with a CFT.

We are most interested in extracting the subleading contributions to $S_A$. Specifically, we study the 2nd Rényi entropy $S_A^{(2)}$ obtained from QMC simulations of 2d spin models. Recent algorithmic advances in QMC calculations of Rényi EE [8,9] make it possible to study the scaling behavior of EE over a sufficiently large subregion [8–16]. In particular, the data quality has reached the level that is needed for the purpose of extracting *subleading* corrections to the "perimeter law" scaling. However, even with the advanced algorithm, since $S_A$ is always dominated by the leading perimeter law, it is still a challenge to reliably extract the subleading contributions. Direct fitting with the leading perimeter law can indeed reveal some information about the subleading contributions with finite system size, but it usually takes enormous effort for data analysis.

To overcome this complexity, in this work, we investigate the following "subtracted EE" [17]:

$$S^s(l) = S_A(2l) - 2S_A(l), \tag{2}$$

where $S_A(2l)$ and $S_A(l)$ are the EE for subregions $A$ with size $2l$ and $l$ with exactly similar shapes. The advantage of the subtracted EE $S^s(l)$ is that the leading perimeter law scaling is supposed to be canceled out, which exposes the logarithmic correction (if exists) as the leading contribution:

$$S^s(l) = s \ln \frac{l}{\epsilon} + O(1/l). \tag{3}$$

We will show that the subtracted EE makes the subleading logarithmic contribution much more explicit for various cases considered in this work. We also compare the results from the subtracted EE and those from direct analysis, including the leading perimeter law scaling.

We will systematically study the scaling of EE in several representative spin models. One of our main objectives is to establish reliable and systematic methods to extract subleading contributions, with the subtracted EE as the main tool for data analysis, and examine finite-size

effects using both the new method based on subtracted EE and the more traditional fitting method. We study logarithmic corrections to EE (or its absence) in the Néel order and the O(3) Wilson-Fisher QCP and find results consistent with expectations from theory. For the Néel order, we find a logarithmic correction whose coefficient is consistent with the theoretical prediction; for the O(3) Wilson-Fisher QCP, we find that there is indeed a logarithmic correction when the boundary of the subregion $A$ has sharp corners, and our data suggests that the logarithmic correction is absent for a smooth boundary cut, to the best of our efforts. We then apply the method to analyze EE in J-Q-type spin models, which realizes the putative DQCP between the Néel and valence-bond solid (VBS) orders [18,19], and demonstrate that the EE exhibits logarithmic corrections even when the boundary of the entangling region is completely smooth without any corners, which is in sharp contrast to the theoretical expectation of a CFT. In addition, the coefficient of the logarithmic correction $s \approx -0.23$ in the J-$Q_3$ model is very close to the one extracted from a square region with four corners without adding a pinning field [10], though the analysis in Ref. [10] was obtained using direct fitting to (1) instead of using the subtracted EE.

We note that the same "subtracted" quantity which reveals the desired subleading contribution can be devised for other quantities such as the "disorder operator," where a subleading logarithmic contribution is also expected at a $(2+1)d$ CFT [20–27].

## 2 Models

The von Neumann and the Rényi entanglement entropy are defined as follows. Let $A$ be a subregion of the system, and $L_A$ is the linear size of $A$, shown in Fig. 1. The von Neumann entanglement entropy $S_A^{\text{vN}}$ is defined as

$$S_A^{\text{vN}} = -\text{Tr}\,\rho_A \ln \rho_A, \tag{4}$$

and the $n$-th Rényi entropy $S_A^{(n)}$ is defined as

$$S_A^{(n)} = \frac{1}{1-n} \ln \text{Tr}\,\rho_A^n, \tag{5}$$

where the $\rho_A$ is the reduced density matrix of the concerned quantum many-body system $\rho_A = \text{Tr}_{\bar{A}}\{e^{-\beta H}\}$ where $\bar{A}$ is the complement of $A$. In all cases we are interested in, the leading contribution of $S_A$ (both von Neumann and Rényi) is proportional to the linear size $L_A$ when $L_A$ is large. Since the purpose of this work is to reliably extract the subleading correction to the EE, the coefficient of the perimeter-law term, which depends on microscopic details, is not of interest to us.

We will analyze three different models whose subleading contribution to the EE is best manifested in the "subtracted EE" $S^s(l)$ defined in Eq. (2).

**(1).** A 2d spin-1/2 model on a square lattice with antiferromagnetic nearest neighbor interaction and ferromagnetic 2nd neighbor interaction. The ground state of this model is known to be the Néel order [9]

The Néel order of a SO(3) invariant system has two Goldstone modes. A phase with Goldstone modes is expected to have the following scaling of the Renyi EE [7]:

$$S_A(l) = al - (s_G + s_{\text{corner}}) \ln l + O(1/l). \tag{6}$$

Here, the logarithmic correction contains two distinct contributions. $s_G$ is "topological", in the sense that it is independent of the shape of $A$ and is fixed by the number of Goldstone modes:

$$s_G = -\frac{n_G}{2}. \tag{7}$$

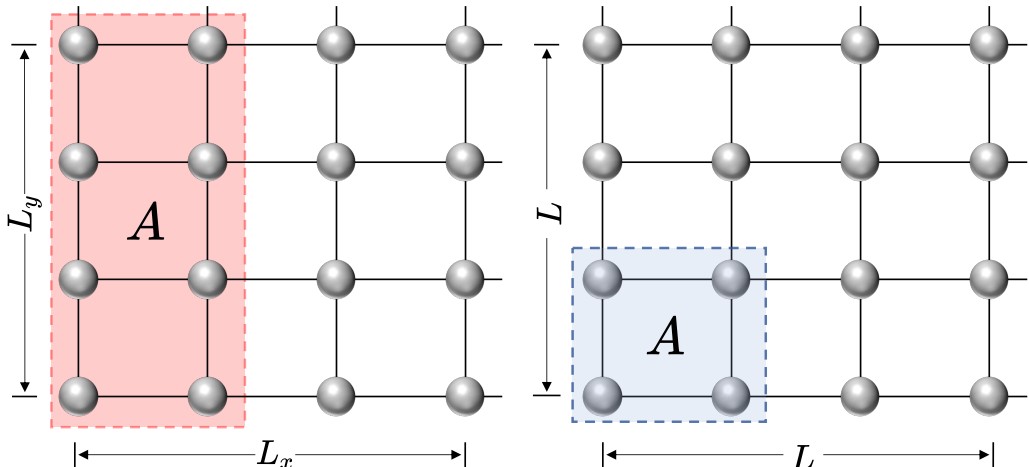

Figure 1: **Illustrations of the entanglement subregions.** The left panel shows an entanglement region $A$ (red shaded area) with smooth boundaries. The right one illustrates a square entanglement region $A$ (blue shaded area) with four $\pi/2$ corners. The lattice is periodic in both dimensions.

This contribution has been verified numerically in Refs. [11, 28, 29] and examined carefully with large-S calculations [30]. The other term, $s_{\text{corner}}$ is similar to the logarithmic corner correction in CFT, and it has been verified numerically in Refs. [31–34]. Typically, $s_G$ is much larger than $s_{\text{corner}}$. Hence, the total subleading logarithmic correction has an opposite sign compared with that of a CFT, which will be discussed next. In fact, to the best of our knowledge, an ordered phase with Goldstone modes is the only well-established theory in $(2+1)d$ with a *positive* logarithmic subleading correction to the perimeter scaling.

**(2).** The bilayer square lattice Heisenberg model, which under increasing the interlayer Heisenberg coupling, goes through a quantum phase transition described by the $(2+1)d$ O(3) Wilson-Fisher CFT [33, 35, 36].

For a $(2+1)d$ CFT, it is important to distinguish entanglement subregions with smooth boundaries and those with sharp corners on the boundary [3, 37, 38]. When the boundary is smooth (e.g., a circle in the continuum, or the bipartition cut in Fig. 1 (a) on a torus), we expect that

$$S_A(l) = al - \gamma + O(1/l), \tag{8}$$

where $\gamma$ is a constant that depends on the geometric shape of $A$. On the other hand, when the boundary has corners whose opening angles are $\alpha_i$ (e.g. four $\alpha = \frac{\pi}{2}$ corners in Fig. 1 (b)), we expect the scaling in Eq. (1), and the logarithmic correction coefficient takes the following form

$$s = \sum_i s(\alpha_i). \tag{9}$$

Both $\gamma$ and $s(\alpha)$ are universal quantities for the CFT. As already mentioned in the Introduction, we have $s(\pi) = 0$, so the logarithmic term is absent for a smooth boundary, which is consistent with Eq. (8). For a unitary CFT, it can be shown on the general ground that [39–41]

$$s^{\text{vN}}(\alpha) \ge 0, \qquad s^{(n)}(\alpha) \ge 0. \tag{10}$$

**(3).** The "J-Q$_2$" and "J-Q$_3$" model which are expected to realize the "deconfined quantum critical point" (DQCP) between the Néel order and the valence bond solid (VBS) order.

The DQCP separates the Néel order with spontaneous symmetry breaking (SSB) of the SO(3) spin rotation symmetry and the VBS order with SSB of the lattice $C_4$ symmetry. DQCP

was proposed as a potential generic unconventional quantum phase transition as it is impossible within the traditional Landau's paradigm [42]. It was found that a direct transition between the Neel and VBS orders can be realized in a spin-1/2 system (the J-Q model) on a square lattice [18]. As a primary example of the non-Landau phase transitions, the DQCP has been proposed to host a number of remarkable properties, including an emergent SO(5) symmetry [43–47], and various non-perturbative dualities [48, 49], and has become a fruitful confluence point between various numerical, analytical and experimental approaches to quantum magnetism [18, 19, 42–82].

On the other hand, whether the DQCP is indeed a generic continuous transition (rather than a first-order transition) is a long-standing open problem. It is found through extensive QMC simulations that the J-Q model exhibits a direct Néel-VBS transition, which appears to be continuous for the largest system size available [18,19,71]. However, violations of scaling have been observed in correlation functions, and the extracted critical exponents drift significantly with system size [44, 58, 59], casting doubts on the conventional scaling [60] and the continuous nature of the transition. More recently, applications of the non-perturbative conformal bootstrap method reveal serious tension between the numerically measured critical exponents and rigorous bounds imposed by conformal symmetry and unitarity [83–85]. Namely, the observed exponents can not possibly correspond to any unitary conformal field theory with a SO(5) symmetry, while SO(5) singlet operators are all irrelevant, as demanded by the proposal of DQCP. To reconcile the numerical results with theory, a number of scenarios have been put forward, such as pseudo-criticality [48, 86–88] and multi-criticality [70, 76, 89, 90].

Recently, the EE in J-Q-type models at the quantum critical point has been studied using the advanced incremental algorithm [10]. Subregions with sharp corners were considered, and the logarithmic subleading correction was extracted from direct fitting to Eq. (1), without using the subtracted EE considered in the current paper. The coefficient was found to be *negative*, violating the positivity constraint Eq. (10). However, to identify the origin of the logarithmic correction, one needs to study the full angle dependence of $s$. In this work, we will add an important new piece of information that was missing in the previous study, that is, $s$ for $\alpha = \pi$, i.e. when the entangling subregion has a smooth boundary. We will show that this case also exhibits a logarithmic correction, with the coefficient very close to the one found for regions with corners. Hence, the logarithmic correction found previously at the DQCP should mostly arise from the smooth boundary rather than the corners.

# 3 Numerical method and fitting schemes

We implement the non-equilibrium incremental algorithm [8–11] for the 2nd Rényi EE calculation in large-scale QMC simulations for quantum spin system under the framework of stochastic series expansion(SSE) QMC [91, 92]. In our algorithm, we calculate the second Rényi EE $S_A^{(2)} = -\ln \text{Tr} \rho_A^2$ which can be re-expressed as $-\ln \frac{Z_A^{(2)}}{Z^{(2)}}$ where $Z^{(2)}$ is two independent replicas in SSE QMC's configuration space and $Z_A^{(2)}$ is two replicas with region $A$ glued together for the boundary condition in imaginary time. The ratio is related to the free energy difference between the two systems represented by the two partition functions by $\Delta F = -\ln \frac{Z_A^{(2)}}{Z^{(2)}}$. The non-equilibrium incremental algorithm measures this quantity by designing a non-equilibrium process from $Z^{(2)}$ to $Z_A^{(2)}$ and calculates the total work ($W$) done during the process. Although in general for a non-equilibrium process, $W \geq \Delta F$, according to Jarzynski's equality [93], the two quantities can be related by $\Delta F = -\ln \left( \left\langle e^{-\beta W} \right\rangle \right)$ where $\langle \cdots \rangle$ denotes the average over many independent non-equilibrium processes. Furthermore, we also improve the speed of this protocol by splitting the non-equilibrium process into many smaller processes, each of which

is conducted by a separate CPU. Combining all these efforts, we are able to measure the EE with very high precision and large enough system sizes to analyze its subleading corrections to area law. More details of the implementation are given in our previous methodology references [11]. We also note that the latest developments of the algorithm have also enabled similar EE computations both in incremental SWAP operator [94] and in interacting fermion systems [12–14, 16].

In the following, we present analysis of the 2nd Rényi entropy $S_A^{(2)}$ in three examples. The models are defined on a square lattice of $L_x \times L_y$ with periodic boundary conditions (Fig. 1). We will consider two types of subregion $A$:

1. a square subregion of size $L_x/2 \times L_y/2$ with four corners,

2. a subregion of size $L_x/2 \times L_y$ with a smooth boundary.

We shall denote the linear size of the entangling subregion by $l$. We are interested in the logarithmic corrections to the perimeter law scaling. The most important improvement compared with previous studies is that, we will investigate a new quantity dubbed "subtracted EE" $S^s(l) = S_A^{(2)}(2l) - 2S_A^{(2)}(l)$, and examine its scaling versus $\ln(l)$ and $1/l$. The subtraction cancels out the leading perimeter law term in the EE and explicitly reveals the nature of the subleading term. If the EE data follows Eq. (1), then we should have

$$S^s(l) = s \ln l - c. \tag{11}$$

As we shall see, the logarithmic subleading term, if it does exist, will become evident from $S^s(l)$.

While $S^s(l)$ allows direct access to the subleading corrections, we will also consider direct fitting to

$$S^{(2)}(l) = al - s \ln l + c, \tag{12}$$

as in this way, we can make use of all the available data points in the fitting. To expose the subleading correction, we rewrite Eq. (12) as

$$\frac{S^{(2)}(l)}{l} = a + \frac{c}{l} + s\frac{\ln(1/l)}{l}, \tag{13}$$

and plot $\frac{S^{(2)}(l)}{l}$ versus $1/l$. In this way, the fitting function becomes $y = sx \ln x + cx + a$, and now it is evident that the logarithmic correction $sx \ln x$, if it exists, would dominate the $cx$ term for small $x$.

Graphically, because $\frac{d^2y}{dx^2} = \frac{s}{x}$, the function is strictly convex/concave for positive/negative value of $s$, which can be inferred from the plots. For a CFT with smooth boundaries, as a log-correction is not expected, in the ideal case we should observe that the slope of the curve converges to $c$ as $x$ (or $1/l$) approaches 0, without any obvious convex/concave feature.

To evaluate the finite size corrections, we will take the following strategies:

1. When we perform the curve-fitting on the data for different system sizes, we keep the largest system sizes fixed and progressively exclude the data for smaller system sizes in the fitting process. We denote the smallest system size retained in the fitting process as $L_{min}$. We examine how $s$ changes as $L_{min}$ is increased and consider the last stable fitted value of $s$ (with largest $L_{min}$ and controllable error bar) as the reference value with minimized finite-size effects. This procedure is performed on the curve fitting of both the $\frac{S^{(2)}(l)}{l}$ versus $1/l$ data and the subtracted EE data.

2. We consider the possibility that the dominant subleading term is not logarithmic, but a $1/l$ term from pure finite size correction. That is, $S^{(2)}$ takes the form

$$S^{(2)}(l) = al + c + \frac{b}{l}, \tag{14}$$

or in terms of the subtracted EE:

$$S^s(l) = -\frac{3b}{2l} - c. \tag{15}$$

In Sec. 4.4, we will compare the quality of fitting to the data between Eq. (11) and Eq. (15). The quality of fitting is evaluated by chi-squared value per degree of freedom $\chi^2/k$ (where the total number of degrees of freedom $k$ equals the number of data points minus the number of fitting parameters). If Eq. (11) already has better performance than Eq. (15), then it suggests that the log-correction better fits the data than the $1/l$-form finite-size corrections. On the contrary, if Eq. (15) fits better to the data, then it suggests that the logarithmic correction is actually absent.

In all the models studied in this work, we find that, except for the O(3) CFT with smooth boundaries, Eq. (11) fits better than Eq. (15), according to their chi-squared values. Most importantly, contrary to the O(3) CFT, we find that at the DQCP with *smooth boundary* for both the J-Q$_2$ and J-Q$_3$ models, there is a clear logarithmic correction to the perimeter-law scaling, with the available system sizes. And the coefficient of the logarithmic correction seems model-dependent, i.e. it is different between the J-Q$_2$ and the J-Q$_3$ model.

## 4 Numerical results

### 4.1 The (2+1)$d$ Néel phase

We compute the Rényi EE for a spin-1/2 model on a square lattice with both antiferromagnetic nearest neighbor interaction, as well as an extra 2nd neighbor ferromagnetic coupling, whose ground state is the Neel order. The Hamiltonian is written as

$$H = J \sum_{\langle i,j \rangle} \mathbf{S}_i \cdot \mathbf{S}_j - J_2 \sum_{\langle\langle i,j \rangle\rangle} \mathbf{S}_i \cdot \mathbf{S}_j, \tag{16}$$

where $\langle i, j \rangle$ and $\langle\langle i, j \rangle\rangle$ label the nearest-neighbor and 2nd neighbor bonds on the square lattice. For the standard Heisenberg model ($J_2 = 0$), the Rényi EE suffers from a strong finite-size effect. However, adding a 2nd neighbor FM coupling to the system is found to enhance the Néel order and significantly reduces the finite size effects in the EE data [9].

We thus perform the simulations at $J = J_2 = 1$ on a $L/2 \times L$ torus with entangled subregion A chosen to be a $L/2 \times L/2$ cylinder, and $l = L$. Our EE results shown in Fig. 2 are obtained from system sizes of $L = 8, 12, \ldots, 56$ and $\beta = L$. Since the ground state of the model spontaneously breaks the spin rotation symmetry with $n_G = 2$ Goldstone modes and there are no sharp corners on the entanglement boundary, according to Eq. (6) and (7), we expect to obtain a subleading $\ln l$ correction to the perimeter law with $s = -1$.

As shown in Fig. 2, we perform the fitting of both $S_A^{(2)}/l$ versus $1/l$, as well as $S^s(l)$ versus $\ln l$. We gradually exclude the data of smaller system sizes in the fitting process. The fitting results are shown in the inset of Fig. 2 (a) and Fig. 2(c), respectively, and we observe that the fitted values of $-s$ concerning the smallest system size $L_{\min}$ gradually converge to the expected value of 1 within error bars for both fitting strategies.

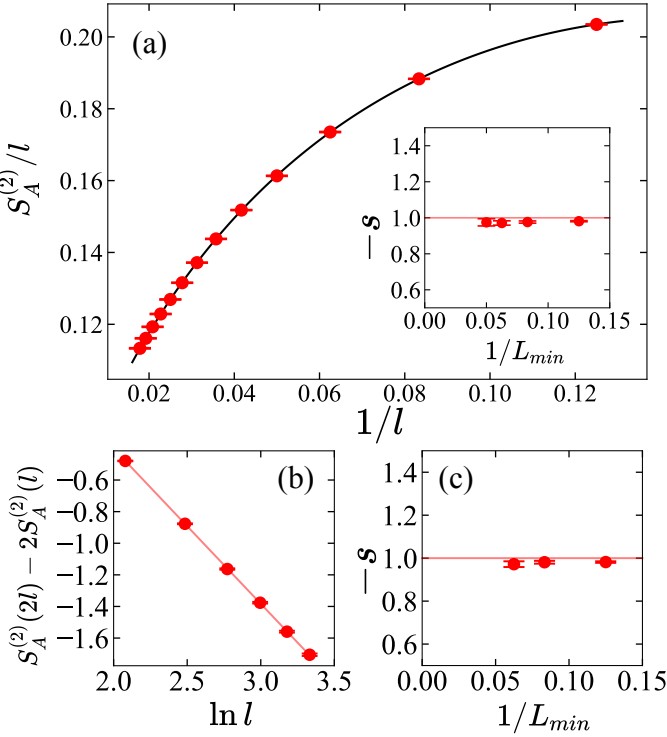

Figure 2: **The second Rényi entropy versus boundary length in the Néel order with smooth boundaries.** (a) $S_A^{(2)}/l$ versus $1/l$ for different boundary lengths. The black line is fitted from Eq. (12). The inset shows the fitted $s$ with respect to the smallest retained system size $1/L_{\min}$ in the fitting process, and the red line denotes the expected results of $-s = \frac{N_G}{2} = 1$. (b) The subtracted EE $S_A^{(2)}(2l) - 2S_A^{(2)}(l)$ versus $\ln(l)$ for different boundary length. The slope of the fitted red line in (b) indicates the log-coefficient $s$. (c) The fitted $s$ from (b) with respect to the smallest retained system size $1/L_{\min}$ in the fitting process.

Furthermore, by comparing the fitting of subtracted EE with respect to $\ln l$ (Fig. 2(b) ) and $1/l$ (Fig. 5(a)) respectively, it is evident that the subtracted EE fits linearly with $\ln l$ rather than $1/l$. Besides visual comparison, we will further examine the fitting quality of the two cases according to their chi-squared values in Sec. 4.4.

## 4.2 (2+1)d O(3) QCP in bilayer antiferromagnetic Heisenberg model

The $(2+1)d$ O(3) phase transition can be realized in a bilayer Heisenberg model [33, 35, 36] defined on a bilayer square lattice with nearest-neighbor antiferromagnetic intra-layer coupling $J$ and inter-layer coupling $J_\perp$, as shown in Fig. 3 (a). The Hamiltonian is

$$H = J \sum_{\langle i,j \rangle} (\mathbf{S}_{i,1} \cdot \mathbf{S}_{j,1} + \mathbf{S}_{i,2} \cdot \mathbf{S}_{j,2}) + J_\perp \sum_i \mathbf{S}_{i,1} \cdot \mathbf{S}_{i,2}, \tag{17}$$

where $\langle i, j \rangle$ denote the nearest neighbor bonds. We choose $g = J_\perp/J$ as the tuning parameter, and previous studies have shown that the critical point $g_c = 2.5220(1)$ separates the Néel ordered phase from the inter-layer dimer product phase (i.e. the disordered phase), and this transition belongs to the $(2+1)d$ O(3) universality class [23, 95].

In the simulation, we compute the 2nd Rényi EE at $g_c$ on a $L \times L$ square lattice with system sizes $L = 4, 8, 12, \ldots, 56$ and inverse temperature fixed at $\beta = L$. Here, we consider both smooth cuts (i.e. regions with a smooth boundary) and square cuts with corners, as illustrated

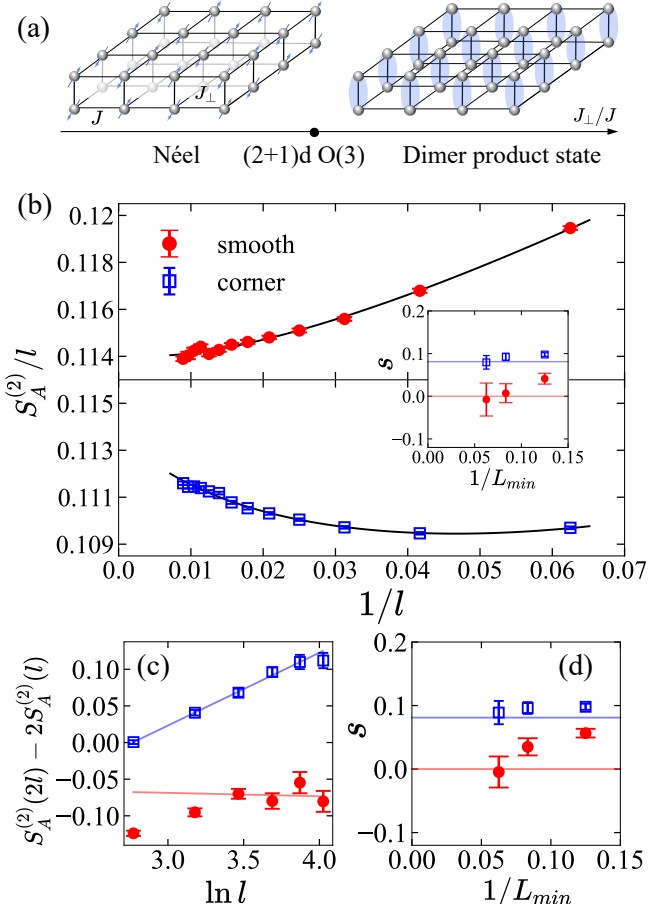

Figure 3: **2nd-order Rényi entropy versus boundary length at the (2+1)$d$ O(3) QCP of bilayer Heisenberg model.** (a) Illustration of the bilayer Heisenberg model, which hosts the (2+1)$d$ O(3) phase transition by tuning the inter-layer coupling strength $J_\perp/J$. (b) EE for both smooth boundary and boundary with corners for different boundary lengths. The black line is fitted from Eq. (12) directly. The inset shows the fitted $s$ with respect to the smallest retained system size $1/L_{\min}$ in the fitting process. (c) The subtracted EE $S_A^s(l)$ versus $\ln l$. The slope of the fitted red line in (c) vanishes at large $l$ within error bars, indicating no log-corrections. Also, the six data points in (c) fit better with a $1/l$ subleading correction rather than a logarithmic correction, based on the $\chi^2/k$ value. (d) The fitted $s$ from data in (c) with respect to the smallest retained system size $1/L_{\min}$ in the fitting process. The red line denotes the expected results for the smooth boundary at QCPs, $s = 0$. The blue line indicates the previously determined log-coefficient with four $\pi/2$ corners at the $(2+1)d$ O(3) QCP.

in Fig. 1 (a) and (b). In both cases, the boundary length of the entanglement region $A$ is $l = 2L$. As shown in Fig. 3 (b), there is a visible difference between the EE scaling behavior for smooth and square cuts. The more obvious convex curve for EE for the square cut suggests a subleading logarithmic correction with a positive $s$.

The subtracted EE $S^s(l)$ vs $\ln l$ is shown in Fig. 3 (c) For the subregion with corners, $S^s(l)$ scales relatively linearly in $\ln l$, with a slope $s \approx 0.098$ if all data points are used in the fitting. In addition, the fitted $s$ remains approximately unchanged within errorbars as one increases $L_{\min}$, as shown both in the inset of Fig. 3(b) from direct fitting, and Fig. 3(d) fitted from $S^s$ vs $\ln l$ in Fig. 3 (c).

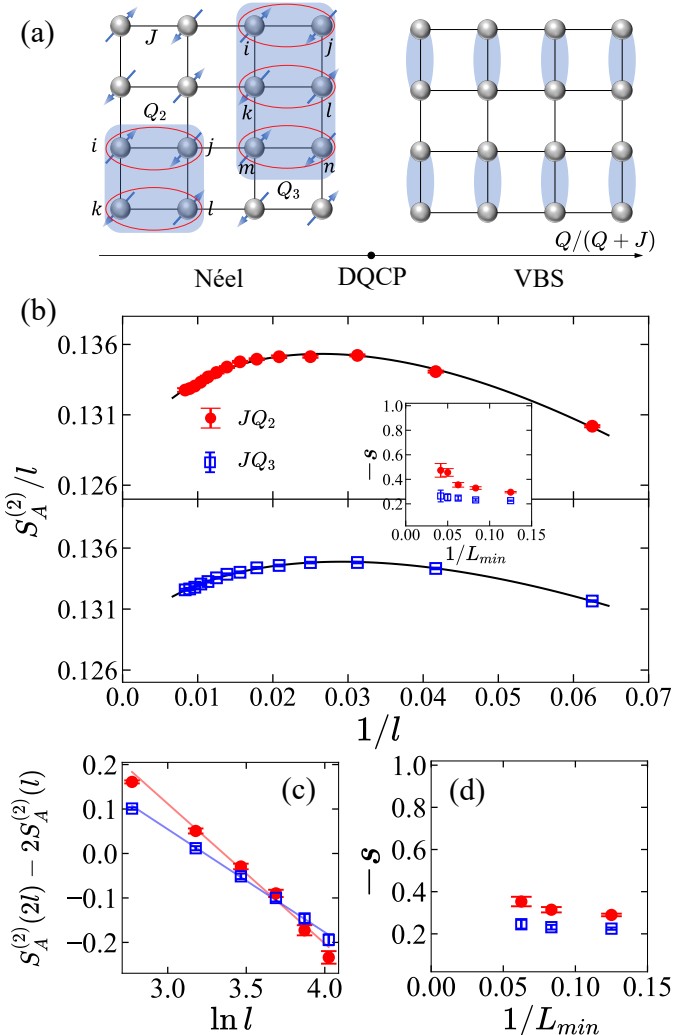

Figure 4: **The 2nd Rényi entropy versus boundary length at the DQCP of the J-Q models with smooth boundaries.** (a) Lattice model with $J$, $Q_2$ and $Q_3$ terms. A DQCP separates the Néel phase and VBS phase. (b) Scaling of EE against boundary length $l$ in both J-Q$_2$ and J-Q$_3$ models with smooth boundaries. The black line is the fitted line of Eq. (12). The inset shows the fitted $s$ concerning the smallest retained system sizes in the fitting process. (c) shows the subtracted EE $S_A^{(2)}(2l) - 2S_A^{(2)}(l)$ versus $\ln l$. The slope of the fitted red line in (c) indicates the log-coefficient $s$. (d) shows the fitted $s$ from (c) concerning the smallest retained system sizes.

In contrast, for the subregion with a smooth boundary, after the first few data points, $S^s(l)$ appears to fluctuate around a constant value, suggesting the absence of a logarithmic correction. Indeed, the fitted $s$ shows significant variations against changing $L_{\min}$, and for the largest allowed $L_{\min}$ we obtained $s = 0$ within error bars. In fact, the behavior of $S_A^{(2)}(l)$ with smooth boundary seems to approach the behavior described by Eq. (8) with large $l$, i.e. there is a constant term $\gamma$ in addition to the perimeter law, though the data points with large $l$ have considerable error bars.

### 4.3 SU(2) DQCP with smooth boundary

This section presents the scaling behavior of the EE with smooth boundaries at the Néel-to-VBS DQCP in the spin-1/2 J-Q models. In Ref. [10], the scaling behavior of the 2nd Rényi entropy for a square region at DQCP of the J-$Q_3$ model with four $\frac{\pi}{2}$ corners has been investigated. If the DQCP is indeed a unitary CFT, the scaling form of the Rényi entropy with an entanglement region $A$ is expected to follow Eq. (12) and the coefficient $s$ of the logarithmic correction arising from sharp corners must be positive. In Ref. [10], $s$ is found to be negative if we try fitting the data with the form Eq. (12). However, since Ref. [10] only analyzed data for angle $\pi/2$, it is unclear whether the fitted $s$ can be interpreted as corner contributions. To clarify this issue, in this section, we consider the Rényi EE at the DQCP for a subregion with smooth boundaries.

We measure the 2nd Rényi EE $S^{(2)}$ in both the J-$Q_2$ and J-$Q_3$ model with the following Hamiltonians

$$
\begin{aligned}
H_{\text{J-}Q_2} &= -J \sum_{\langle ij \rangle} P_{i,j} - Q \sum_{\langle ijkl \rangle} P_{ij} P_{kl} \,, \\
H_{\text{J-}Q_3} &= -J \sum_{\langle ij \rangle} P_{i,j} - Q \sum_{\langle ijklmn \rangle} P_{ij} P_{kl} P_{mn} \,,
\end{aligned}
\tag{18}
$$

where $P_{ij} = \frac{1}{4} - \mathbf{S}_i \cdot \mathbf{S}_j$ is the two-spin singlet projector, and the ground state of the $Q$ term is a valance-bond-solid (VBS) state, as shown in Fig. 4 (a). The Néel-to-VBS phase transition occurs at $Q/(J+Q) = 0.59864$ for the J-$Q_3$ model [23], and at $Q/(J+Q) = 0.961$ for the J-$Q_2$ model [19].

We perform simulations on $L \times L$ square lattices with sizes $L = 8, 12, 16, \ldots, 60$ at the DQCPs of the J-Q models. The entangled subregion $A$ is half of the torus, a cylinder with smooth boundaries of length $l = 2L$. Fig. 4 (b) plots $S_A^{(2)}/l$ against $1/l$ for both models, and the curves are clearly concave, suggesting a negative $s$. The subtracted EE $S^s(l)$ exhibits linear scaling against $\ln l$ as illustrated in Fig. 4 (c), in contrast with its apparent non-linear behavior with $1/l$ as shown in Fig. 5 (c). The fitted slope of Fig. 4 (c) with all available data points is found to be $s = -0.224(5)$ for the J-$Q_3$ model and $s = -0.289(6)$ for the J-$Q_2$ model, see Table. 1. We have also examined the effect of changing $L_{\min}$. As shown in the inset of Fig. 4(b) and Fig. 4 (d), $s$ for the DQCP in the J-$Q_3$ model seems stable against $L_{\min}$, while the one for the J-$Q_2$ model drifts slightly as $L_{\min}$ increases. Our results show that even for subregions without sharp corners, within the available system size, the scaling of Rényi EE at the Néel-to-VBS DQCP still has a logarithmic correction to the leading perimeter law scaling, with a *positive* coefficient. It is also worth noting that the J-$Q_2$ and J-$Q_3$ give different fitted values of $s$, suggesting a model-dependent $s$ for DQCP.

### 4.4 Quality of fitting analysis

In this section, we study the quality of the fitting. One quantitative way of measuring the quality of the data fitting to a model is through the $\chi^2$ value per degree of freedom. Suppose we have data points $(x_i, y_i), i = 1, 2, \ldots, N$ fitted to a model $y = f(x)$ with $r$ fitting parameters, then $\chi^2$ is defined as

$$
\chi^2 = \sum_{i=1}^{N} \frac{(f(x_i) - y_i)^2}{\sigma_i^2} \,.
\tag{19}
$$

Here $\sigma_i$ is the uncertainty of data $y_i$. $k = N - r$ is the effective number of degrees of freedom. If the data were indeed given by the fitting function with random errors, then one expects that for sufficiently large $k$, $\chi^2/k$ should be close to 1. More precisely, $\chi^2/k$ should be typically distributed within the range $[1 - \sqrt{2/k}, 1 + \sqrt{2/k}]$. A larger value of $\chi^2/k$ suggests underfitting, and a smaller value of $\chi^2/k$ does not necessarily imply good fitting but can also

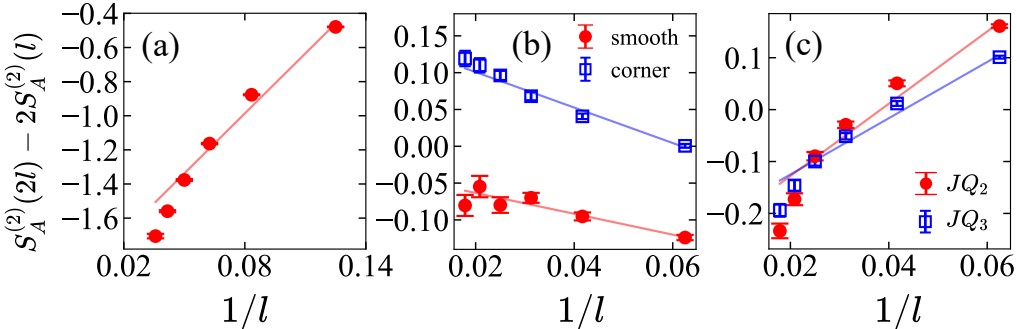

Figure 5: **The fitting of the subtracted EE $S_A^{(2)}(2l) - 2S_A^{(2)}(l)$ v.s. $1/l$.** (a) The subtracted EE of the Néel phase for subregion A with a smooth boundary; (b) the O(3) QCP with both smooth boundary and boundary with corners; (c) the DQCP of the J-Q models with smooth boundary. The fits in these three panels have lower quality compared to their counterparts in Fig. 2 (b), Fig. 3 (c), and Fig. 4 (c), except for the smooth boundary case of O(3) QCP, which fits better with a $1/l$ subleading correction. The corresponding $\chi^2/k$ values are listed and compared with the $\ln l$ fittings in Table 1.

suggest overfitting or problematic uncertainties in the data [96, 97]. For our problem, we always have $k = 4$ for the analysis of subtracted EE, so $\chi^2/k$ is typically distributed within $[1 - 1/\sqrt{2}, 1 + 1/\sqrt{2}] \approx [0.3, 1.7]$ for a good fit.

We compare the fitting of the subtracted EE with the $\ln l$ form to that with a $1/l$ form to identify whether the subleading correction we observe is truly logarithmic. We fit the subtracted EE, $S^s(l)$ with respect to linear functions of $\ln l$ and $1/l$, respectively, and compare their fitting qualities. We list the obtained $\chi^2/k$ values and the fitted subleading coefficients in Table 1. We can see that the case of the $(2+1)d$ O(3) QCP with a smooth boundary is an outlier, as it is the only case whose subtracted EE data fits better with $1/l$, i.e. Eq. (15).

## 5 Discussions

In this work, we study subleading corrections to the EE for several different models. We introduce the quantity "subtracted EE," which cancels out the leading perimeter law contribution and explicitly reveals the subleading correction. We demonstrate that the subtracted EE gives the desired universal logarithmic correction in several cases, including the Goldstone phase and the O(3) CFT, where the subregion has sharp corners. We also found that the subtracted EE at the DQCP scales linearly with $\ln l$ for the available system sizes.

Here, we discuss one possible explanation for the observed $\ln l$ scaling at the DQCP. Since our numerical simulations are done on finite systems, it is important to further understand finite-size corrections to the formula. Theoretically, this question has been studied in the Goldstone phase, and the EE with finite-size corrections takes the following form [7, 29]:

$$S_A(L) = aL + \frac{n_G}{2} \ln \left[ \left( \rho_s(L)L \right)^{1/2} \left( I(L)L \right)^{1/2} \right] + c \,. \tag{20}$$

Here $\rho_s(L)$ and $I(L)$ are the finite-size value of the spin stiffness and the transverse spin susceptibility, respectively. Following conventional finite-size scaling we can expand $\rho_s(L) = \rho_s + \frac{u}{L} + \cdots$ and similarly $I(L) = I + \frac{v}{L} + \cdots$, and for $L \gg \frac{u}{\rho_s}, \frac{v}{I}$ one recovers

$$S_A(L) = aL + \frac{n_G}{2} \ln L + c' + O\left(\frac{1}{L}\right) \,. \tag{21}$$

Table 1: **Obtained subleading coefficients and $\chi^2/k$ values.** The second and third columns present the fitted log-coefficient with $L_{\min} = 8, 16$, respectively. Error bars in the parentheses represent the uncertainty in the last significant digit, e.g., $-0.004(24)$ means $-0.004 \pm 0.024$. The last two columns list the $\chi^2/k$ values for the fitting of the subtracted EE data with $L_{\min} = 8$, using Eq. (3) ($\ln l$ correction) and Eq. (15)($1/l$ correction), respectively.

| | Fitted $s$ with $L_{\min} = 8$ | Fitted $s$ with $L_{\min} = 16$ | $\chi^2/k$ | |
| --- | --- | --- | --- | --- |
| | | | $\ln l$ | $1/l$ |
| Néel, smooth | $-0.982(3)$ | $-0.97(1)$ | 0.31 | 481.9 |
| O(3), smooth | $0.056(7)$ | $-0.004(24)$ | 2.38 | 1.30 |
| O(3), corner | $0.098(4)$ | $0.088(18)$ | 0.23 | 2.66 |
| J-Q$_2$, smooth | $-0.289(6)$ | $-0.35(2)$ | 3.38 | 25.2 |
| J-Q$_3$, smooth | $-0.224(5)$ | $-0.24(2)$ | 0.49 | 13.1 |

However, if $\rho_s(L)$ and $I(L)$ have unconventional finite-size scaling behavior, e.g. $\rho_s(L)L \sim I(L)L \sim L^x$ with $x \neq 1$, then the coefficient of the logarithmic correction will instead becomes $\frac{n_G}{2}x$. This kind of unconventional finite-size scaling has been indeed observed in the J-Q$_2$ model [98] where both $\rho_s(L)L$ and $I(L)L$ scale as $L^{0.285}$ at the DQCP. It would then give $s = -0.285$ (for $n_G = 2$) when $L$ is big enough, which is close to the value found in this work. It would be worth investigating this picture further in the future, as well as other possible explanations of the logarithmic correction.

In this paper, we have focused on the logarithmic subleading corrections. For a real $(2+1)d$ CFT with a smooth boundary, the subleading correction to the perimeter law is a constant $\gamma$, which depends on the geometric shape of the global spatial manifold, as well as the subregion. In the O(3) CFT example, we have already discussed in Sec. 4.2 that for the smooth boundary case our data analysis suggests that a logarithmic correction is absent, and if we fit the data with Eq. (15) we find $\gamma \approx -0.04 \pm 0.01$ with $L_{\min} = 8$ (see Fig. 5 (b)), and $\gamma \approx -0.07 \pm 0.03$ with $L_{\min} = 16$, which is summarized in Table. 2. Note that this constant correction was found to be $\gamma \approx 2.25$ for the free O(3) scalar theory [99] on the same geometric set-up, which differs significantly from our numerical result, while it is usually expected that the O(3) Wilson-Fisher fixed point is close to the free boson fixed point. This obvious difference warrants further theoretical and numerical analysis in the future.

Table 2: **Fitted $\gamma$ in Eq. (8) using Eq. (15) at O(3) QCP with smooth cut.** The second, third, and fourth columns present the fitted constant term $\gamma$ with $L_{\min} = 8, 12, 16$, respectively.

| | $L_{\min} = 8$ | $L_{\min} = 12$ | $L_{\min} = 16$ |
| --- | --- | --- | --- |
| Fitted $\gamma$, O(3) smooth | -0.04(1) | -0.04(2) | -0.07(3) |

# Acknowledgments

We thank Jonathan D'Emidio, William Witczak-Krempa, Yuan Da Liao, Yin-Chen He, Yan-Cheng Wang, and Anders Sandvik for valuable discussions. We acknowledge the readers to other two related works recently preprinted online by other authors [100, 101].

**Funding information** MHS, JRZ and ZYM acknowledge the support from the Research Grants Council (RGC) of Hong Kong Special Administrative Region of China (Project Nos. 17301721, AoE/P-701/20, 17309822, HKU C7037-22GF, 17302223), the ANR/RGC Joint Research Scheme sponsored by RGC of Hong Kong and French National Research Agency (Project No. A_HKU703/22). We thank HPC2021 system under the Information Technology Services at the Department of Physics, University of Hong Kong as well as the Beijng PARATERA Tech CO.,Ltd. (URL: https://cloud.paratera.com) for providing HPC resources that have contributed to the research results reported within this paper.

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
