# Peer review of "Extracting subleading corrections in entanglement entropy at quantum phase transitions"

_SciPost Physics_

## Round 2 · Referee Report · Anonymous (Referee 1) · 2024-3-26

Strengths

  1. The paper presents state-of-the-art numerical simulations of the finite-size scaling of the Rényi entropy for different quantum many-body phases.

  2. Their simulations cover a broad range of cases, including deconfined quantum critical points (DQCP), which have recently been under debate in the literature.

3.Their precise numerical results allow not just the confirmation of some theoretical results, but also the identification of possible discrepancies with theoretical expectations.

  1. The authors even propose some possible explanations for the size scaling of the Rényi entropy in the case of DQCP.

Weaknesses

  1. The authors could comment more on some details of the QMC simulations used in this work.

  2. The idea of using the "subtracted EE" was already explored in similar contexts (e.g., in New Journal of Physics 22 (1), 013044); they could have commented on this

Report

The paper by Menghan Song et al. employs recent algorithmic advances to obtain the finite-size scaling of the Renyi entropy in different paradigmatic quantum phases. In particular, their large-scale simulations provide access to universal subleading corrections to the Renyi entanglement entropy (EE).

Overall, despite the methods employed by the authors not being new, I believe their precise results for the universal corrections of the Rényi EE are relevant to the field. Specifically, their results allow for drawing comparisons with theoretical expectations from quantum field theories. For example, the numerical results for the O(3) Quantum Critical Point (QCP) indicate a much smaller constant than expected for the free boson fixed point, while in the DQCP, they observe a positive coefficient for the logarithmic correction.

Requested changes

I just have suggestions of change.

  1. Include the results of the constant of constant $\gamma$ in table Table 1. The absolute value $\gamma$ increases with Lmin? Maybe the author could mention the its value for a third value of Lmin, similar to what they do for the content $s$.

  2. Include some information about the QMC simulations.

  • validity: high
  • significance: high
  • originality: ok
  • clarity: good
  • formatting: good
  • grammar: good

Author:  Menghan Song  on 2024-05-27  [id 4516]

(in reply to Report 1 on 2024-03-26)

We would like to thank Referee A for his/her review of our work.

Regarding the comments, we respond one by one below and make corresponding changes in the revised manuscript.

Comment 1: The idea of using the "subtracted EE" was already explored in similar contexts (e.g., in New Journal of Physics 22 (1), 013044); they could have commented on this.

Reply 1: We are grateful for the referee's attention to pertinent literature. We were indeed not aware of it. We have incorporated this reference into our revised manuscript.

Comment 2: The authors could comment more on some details of the QMC simulations used in this work.

Reply 2: We thank the referee for the suggestion and have added more discussions on the implementation of the QMC simulation in the revised manuscript.

Comment 3: Include the results of the constant of constant
$\gamma$ in table Table 1. The absolute value $\gamma$
increases with $L_{min}$? Maybe the author could mention its value for a third value of $L_{min}$, similar to what they do for the content $s$.

Reply 3: We appreciate the referee's helpful suggestions and have included a table in the revised manuscript that summarizes the constant \(\gamma\) fitted at various \(L_{\min}\) values. For all \(L_{\min}\) values, \(\gamma\) remains negative and small. Notably, as \(L_{\min}\) increases from 8 to 16, the absolute value of \(\gamma\) rises from 0.04(1) to 0.07(3).

---

## Round 2 · Referee Report · Anonymous (Referee 2) · 2024-4-26

Strengths

This paper has several strengths

1- Very reliable QMC study of the Entanglement Entropy, a quantity notoriously difficult to measure. 2-Very thorough, careful and important study of various states of 2D quantum matter (symmetry-broken, (2+1)d O(3) QCP, DCQP). 3-Trigger very interesting questions about possible new log corrections at the DCQP

Weaknesses

I do not see any weaknesses

Report

I first want to apologise for my late report...

In this work, the authors present very convincing QMC results for the entanglement entropy in various important quantum states of matter.

Their results are very well presented and very convincing. I have only very minor comments or suggestions that could serve either for this work or for future research.

1/ It may be fair to recall some of the past efforts made using large-S calculations that basically also brought strong justifications to Eq. (6), in particular also to extract subleading constant strip corrections. For example using square vs strip contributions in PHYSICAL REVIEW B 92, 115126 (2015) they precisely extract corner terms, a trick that could also be used by the present authors for their study.

2/ In Fig. 4(b), it seems that the area-law prefactor is the same for both DQCPs : is that true?

3/ Concerning the log correlations at the DQCPs, I like very much the interpretation in terms of the unusual finite-size scaling of the stiffness and susceptibilities that may best be written like this S(L)= aL + \frac{n_G}{2}\ln\left[L\sqrt{\chi_\perp\rho_s}\right]+\gamma_{ord} (see Eq. 1.5 from Metliski and Grover).

From the above form, is that correct, following "Quantum criticality with two length scales" by Shao, Guo and Sandvik, that one expects the log term to be simply \frac{n_G}{2}(1-\nu/\nu')\ln L ? If so could the authors maybe write it and comment why this correction is different between the two DQCP models?

4/ Finally do the authors expect similar unconventional corrections at a first order transition between a Néel order and a gapped state for instance?

Despite these few questions and comments I recommend this work to be published.

Recommendation

Publish (easily meets expectations and criteria for this Journal; among top 50%)

  • validity: top
  • significance: top
  • originality: high
  • clarity: high
  • formatting: excellent
  • grammar: excellent

Author:  Menghan Song  on 2024-05-27  [id 4515]

(in reply to Report 2 on 2024-04-26)

We would like to thank referee B for his/her high assessments of our work. We respond to the comments below and make corresponding changes in the revised manuscript.

Comment 1: It may be fair to recall some of the past efforts made using large-S calculations that basically also brought strong justifications to Eq. (6), in particular also to extract subleading constant strip corrections. For example using square vs strip contributions in PHYSICAL REVIEW B 92, 115126 (2015) they precisely extract corner terms, a trick that could also be used by the present authors for their study.

Reply 1: We appreciate the referee's consideration of relevant literature. This reference has been integrated into our revised manuscript accordingly.

Comment 2: In Fig. 4(b), it seems that the area-law prefactor is the same for both DQCPs : is that true?

Reply 2: We thank the referee for raising this interesting question. The fitted area-law prefactor for both the \(JQ_2\) and \(JQ_3\) models is close to \(\sim 0.13\), derived using all data points in curve fitting for \(L_{\min} = 8\). But when we gradually increase the $L_{\min}$, they start to deviate and that of \(JQ_2\) drifts slightly away from that of \(JQ_3\). But the phenomenon that the two models have very similar perimeter-law coefficients is indeed a feature that we have not noticed before. We must admit that we do not fully understand this feature, and we plan to further study this in the future.

Comment 3: Concerning the log correlations at the DQCPs, I like very much the interpretation in terms of the unusual finite-size scaling of the stiffness and susceptibilities that may best be written like this
$S(L)= aL + \frac{n_G}{2}\ln\left[L\sqrt{\chi_\perp\rho_s}\right]+\gamma_{ord}$ (see Eq. 1.5 from Metliski and Grover). From the above form, is that correct, following "Quantum criticality with two length scales" by Shao, Guo and Sandvik, that one expects the log term to be simply $\frac{n_G}{2}(1-\nu/\nu')\ln L$ ?
If so could the authors maybe write it and comment why this correction is different between the two DQCP models?

Reply 3: Yes, this is exactly what we mean. In fact we have used ``$x$" rather than $1-\nu/\nu'$ in our manuscript. As for the later question, because there are many subtleties in the $JQ_2$ and $JQ_3$ models, $\nu/\nu'$ might be different for the two models and this may explain why the log coefficient is different for the two cases. We note that this is only a possible scenario, and as there are not enough numerical data on the two length scale behavior on $JQ_3$ model, we choose to not discuss this difference too much in our manuscript and leave it for further investigations.

Comment 4: Finally do the authors expect similar unconventional corrections at a first order transition between a Néel order and a gapped state for instance?

Reply 4: We appreciate the referee for raising this intriguing question. To the best of our knowledge, no prior studies discuss the scaling behavior of entanglement entropy (EE) at first-order transitions. However, we have a simple argument that for an absolute and strong first-order quantum phase transition between two very different phases, the scaling of EE will be dominated by the scaling of EE of the phase, which has a smaller area law coefficient. The argument is as follows.

At a strong 1st order transition, the system is kind of like a classical mixture of two very different states $|1\rangle$ and $|2\rangle$, with some classical probability $p_1$ and $p_2$. Then if we apply the swap operator to the mixture of the two states to calculate the EE, we will get $S^{(2)}_{A}=-\ln(\langle Swap \rangle)\sim -\ln(p_{1} \langle 1|Swap|1\rangle + p_{2} \langle 2|Swap|2\rangle)=-\ln(p_1 e^{-S_{1}} +p_2 e^{-S_{2}})$. Generally EE scale as $S=aL+b\ln(L)+c$, so that the state with a smaller area law coefficient will dominate the scaling of $S^{(2)}_{A}$.

However, for a phase transition with a very weakly first order behavior, the situation becomes much more complicated, as the order parameter is extremely small and the correlation function resembles a real continuous phase transition at small system sizes. In this case, the scaling of EE can be affected by many elements, and we are still exploring a way to understand all these effects.

---

## Editorial Decision

resubmitted